# Glycan modification of antigen alters its intracellular routing in dendritic cells, promoting priming of T cells

Ingeborg Streng-Ouwehand[1*], Nataschja I Ho[2*], Manja Litjens[1], Hakan Kalay[1], Martine Annemarie Boks[1], Lenneke AM Cornelissen[1], Satwinder Kaur Singh[1], Eirikur Saeland[1], Juan J Garcia-Vallejo[1*], Ferry A Ossendorp[2*], Wendy WJ Unger[1*†‡], Yvette van Kooyk[1*†]

[1]Department of Molecular Cell Biology and Immunology, VU University Medical Center, Amsterdam, Netherlands; [2]Department of Immunohematology and Blood Transfusion, Leiden University Medical Center, Leiden, Netherlands

*For correspondence:
i_ouwehand@hotmail.com (ISO); N.I.S.C.Ho@lumc.nl (NIH); jj.garciavallejo@vumc.nl (JJGV); f.a.ossendorp@lumc.nl (FAO); w.unger@erasmusmc.nl (WWU); y.vankooyk@vumc.nl (YvK)

†These authors contributed equally to this work

Present address: ‡Department of Pediatrics, Erasmus University Medical Center, Rotterdam, Netherlands

Competing interests: The authors declare that no competing interests exist.

**Abstract** Antigen uptake by dendritic cells and intracellular routing of antigens to specific compartments is regulated by C-type lectin receptors that recognize glycan structures. We show that the modification of Ovalbumin (OVA) with the glycan-structure Lewis$^X$ (Le$^X$) re-directs OVA to the C-type lectin receptor MGL1. Le$^X$-modification of OVA favored Th1 skewing of CD4$^+$ T cells and enhanced cross-priming of CD8$^+$ T cells. While cross-presentation of native OVA requires high antigen dose and TLR stimuli, Le$^X$ modification reduces the required amount 100-fold and obviates its dependence on TLR signaling. The OVA-Le$^X$-induced enhancement of T cell cross-priming is MGL1-dependent as shown by reduced CD8$^+$ effector T cell frequencies in MGL1-deficient mice. Moreover, MGL1-mediated cross-presentation of OVA-Le$^X$ neither required TAP-transporters nor Cathepsin-S and was still observed after prolonged intracellular storage of antigen in Rab11$^+$LAMP1$^+$ compartments. We conclude that controlled neo-glycosylation of antigens can crucially influence intracellular routing of antigens, the nature and strength of immune responses and should be considered for optimizing current vaccination strategies.

## Introduction

The induction of T cell immunity to viruses or tumors involves the presentation of viral or tumor antigens by antigen-presenting cells (APC) in the context of major histocompatibility complex (MHC) molecules. Loading of exogenously-derived antigens onto MHC class I molecules, a process known as cross-presentation (*Carbone and Bevan, 1990*) is required to activate antigen-specific CD8$^+$ T cells. Furthermore, cognate CD4$^+$ T cell help is important for licensing the APC, which is essential for effective CD8$^+$ T cell priming and induction of long-lasting memory (*Bennett et al., 1997*; *Schoenberger et al., 1998*). The molecular mechanisms underlying cross-presentation have been studied intensively, however, little is still known on the nature of the antigens and stimuli that are required for APC to route exogenous antigens efficiently into the MHC class I presentation pathway.

Most exogenous antigens that originate from tumor cells or viruses are glycosylated in their native form (*Apweiler et al., 1999*). The relative composition of the glycosylation machinery, which includes all the necessary glycosylation-related enzymes and co-factors, determines the final configuration of the glycan structures that decorate *N*- or *O*-linked glycosylation sites present in glycoproteins. The glycosylation machinery can be affected by multiple physio-pathological cues, including proliferation, activation, and the transformation status of the cell (*Ohtsubo and Marth, 2006*), and depends on environmental factors. APC, such as dendritic cells (DCs) and macrophages are able to

**eLife digest** Immune cells called dendritic cells play a crucial role in defending the body against tumor cells and invading viruses. The dendritic cells take up molecules called antigens from these threats and then display them on their surface. This enables the antigens to be identified by other immune cells that are capable of killing the viruses and the tumor cells. The dendritic cells recognize the antigens with the help of receptor proteins called C-type lectin receptors (CLRs). These receptors can bind to sugar molecules that are naturally found on many antigens. For example, a C-type lectin receptor called MGL1 can bind to sugars known as Lewis[X] and Lewis[a] on tumor and virus proteins. However, it is not clear how important these receptors are in triggering immune responses.

An antigen called Ovalbumin – which is found in chicken egg white – can trigger immune responses in mammals and so researchers often use it to study the immune system. Although this antigen has several sugar molecules attached to it, quite a large amount of Ovalbumin is needed to trigger strong immune responses. Now, Streng-Ouwehand et al. examine whether attaching Lewis[X] to Ovalbumin can results in stronger immune responses in mice.

The experiments show that injecting mice with Ovalbumin-Lewis[X] triggers a much stronger immune response than normal Ovalbumin does. This enhanced response was not observed in mice that lacked the MGL1 receptor, which suggests that this receptor is involved in detecting Ovalbumin-Lewis[X]. Furthermore, the dendritic cells store the altered Ovalbumin for longer than they store normal Ovalbumin, which gives the cells more time to present the altered Ovalbumin to other immune cells.

Vaccines and some other therapies help to boost immune responses to viruses and tumors by exposing the body to antigens. However, these therapies often use antigens that don't have sugar molecules attached to them, or are missing the sugar molecules they would normally have. Streng-Ouwehand et al.'s findings suggest that adding specific sugars to antigens in immune therapies might help to make these therapies more effective.

sense glycans exposed on either self- or pathogen-derived antigens via glycan-binding proteins. Amongst these glycan-binding proteins are C-type lectin receptors (CLRs) that recognize defined carbohydrate-structures through their carbohydrate recognition domain (CRD). Depending on the amino acid sequence, the CRD bears specificity for mannose, fucose, galactose, sialylated-, or sulfated structures. CLRs function both as antigen uptake and/or signaling receptors that modify DC-induced cytokine responses thereby influencing T cell differentiation (*van and Rabinovich, 2008*). The specialized internalization motifs in the cytoplasmic domains of CLRs allow the rapid internalization of antigens upon interaction (*Engering et al., 2002*; *Herre et al., 2004*). This suggests that DCs use CLRs to 'sense' the natural glycan composition of tissues and invading pathogens and, in response to this recognition, are able to modulate immune responses (*Geijtenbeek and Gringhuis, 2009*).

The conjugation of antigens to CLRs-specific antibodies, such as DEC205, mannose receptor (MR), Dendritic Cell-Specific Intercellular adhesion molecule-3-Grabbing Non-integrin (DC-SIGN) or CLEC9A, has proven to be an effective way to direct antigens to DCs, resulting in enhanced antigen uptake and presentation on MHC molecules (*Bonifaz et al., 2004*; *Hawiger et al., 2001*; *Caminschi et al., 2008*; *Chatterjee et al., 2012*). Also glycans specific for CLRs have shown their targeting specificity and potential to improve antigen uptake and presentation in MHC class I and II molecules when coupled to antigen formulations (*Singh et al., 2009a*; *Garcia-Vallejo et al., 2013*; *Unger et al., 2012*; *Unger, 2014*; *Aarnoudse et al., 2008*; *Yang et al., 2008*; *Singh et al., 2011*). To achieve immunity rather than tolerance inclusion of a strong adjuvant is necessary. Little is known on how naturally glycosylated antigens or alterations in the glycosylation of antigens may change and re-direct antigen internalization via CLRs, in addition to subsequent processing and presentation.

MR was shown to mediate cross-presentation of the model-antigen ovalbumin (OVA) in a Toll-like receptor (TLR)-dependent manner and to recruit TAP-1 to endocytic organelles (*Burgdorf et al.,*

2007; *Burgdorf et al., 2008*). Importantly, cross-presentation of OVA was only effective using high amounts of antigen (*Burgdorf et al., 2007*; *Burgdorf et al., 2008*). The interaction of OVA with MR, which has specificity for mannose (*Taylor et al., 1992*), was speculated to be dependent on the presence of mannose glycans on OVA (*Burgdorf et al., 2007*). In the current study, we investigated the effect of modifying the glycan composition of OVA on the efficacy of cross-presentation, priming and differentiation of T cells. We hypothesized that modification of the antigen with specific glycans would re-direct antigens to other CLRs resulting in altered intracellular routing and presentation of antigen and Th differentiation. We have chosen to conjugate the carbohydrate structure Lewis$^X$ (Le$^X$) to 2 free cysteine residues within the native OVA glycoprotein. Le$^X$ is a ligand of the murine C-type lectin macrophage galactose-type lectin (MGL)-1 (*Singh et al., 2009b*), which is expressed on murine plasmacytoid DC, CD8$^+$ and CD8$^-$ splenic DC, DC in the small intestines, the sub-capsular and intra-follicular sinuses of T cell areas in lymph nodes, and on DC and macrophages in the dermis of skin (*Dupasquier et al., 2006*; *Tsuiji et al., 2002*; *Denda-Nagai et al., 2010*; *Segura et al., 2010*). Murine MGL1 is one of two homologues (MGL1 and MGL2) of human MGL (huMGL), which has been shown to interact with tumor cells through glycans exposed on MUC1 as well as with various pathogens (*van Vliet et al., 2008*). However, MGL1 and MGL2 bind different glycan structures: while MGL1 binds Le$^X$ and Le$^a$ glycans, MGL2 binds *N*-acetylgalactosamine (GalNAc) and galactose glycan structures (*Singh et al., 2009b*). The YENF motif in the cytoplasmic tail of huMGL is essential for uptake of soluble antigens, which are subsequently presented to CD4$^+$ T cells (*van Vliet et al., 2007*). Murine MGL1 contains a similar motif (YENL) in its cytoplasmic tail, which likely plays a similar and important role in antigen uptake (*Yuita et al., 2005*). Glycan binding to the CLRs DC-SIGN, Dectin-1 and Dectin-2 has been demonstrated to trigger the signaling capacity of these receptors, modulating DC-mediated T helper cell (Th) differentiation and cytokine production by DCs (*Geijtenbeek and Gringhuis, 2009*). This illustrates that glycan epitopes may not only improve antigen presentation, but may also affect Th differentiation and shape specific adaptive immune responses. This is underlined by the observation that DC-SIGN signaling induced upon sensing of specific mannose or fucose structures on pathogens differentially directed Th differentiation (*Gringhuis et al., 2014*).

Here, we demonstrate that conjugation of the Le$^X$ carbohydrate to the model antigen OVA alters its routing from a MR- and TAP1-dependent cross-presentation pathway into a TAP1- and Cathepsin-S-independent pathway, devoid of any TLR-signaling dependence, which simultaneously enhanced CD8$^+$ T cell priming and Th1 skewing of CD4$^+$ T cells. Cross-presentation was associated with prolonged intracellular storage of antigen in Rab11$^+$LAMP1$^+$ vesicles. Our results illustrate that small changes in the glycosylation profile of protein antigens can have a great impact on antigen routing within DCs, affecting cross-presentation and Th cell differentiation.

## Results

### Identification of glycans on native and glycan-modified OVA, and the consequences for CLR-specific binding and cross-presentation

The well-known model antigen OVA carries one *N*-glycosylation site at N293 and, although the variety of OVA glycans have been previously described, their relative abundance has never been characterized in detail. We determined the glycan profile of OVA by normal-phase HPLC coupled with electrospray ionization mass spectrometry with an intercalated fluorescence detector. We found that the majority of glycan species on OVA corresponded to the complex-type (54.9%), while mannose-rich glycans (mainly Man5 and Man6, potential ligands of the MR (*Taylor and Drickamer, 1993*)) represented only 23% of all glycoforms (*Figure 1A*). The remaining glycoforms were of the hybrid (16.2%) and oligomannose (Man3, 3%) type. The presence of mannose rich glycans on OVA correlate with the earlier reports that in particular MR interacted with OVA (*Burgdorf et al., 2006*) and mediated the TLR dependent and TAP-1 dependent cross-presentation of OVA (*Burgdorf et al., 2008*). We reasoned that the conjugation of additional glycans to OVA would alter its CLR-dependent effects. To do this, we conjugated the tri-saccharide glycan structure Le$^X$ (Galβ1–4(Fucα1–3)GlcNAc) to the free cysteins of OVA using standard thiol-maleimide chemistry via the bifunctional crosslinker MPBH. Le$^X$ is a well characterized carbohydrate ligand of the C-type lectin MGL1 (*Hawiger et al., 2001*; *Singh et al., 2009b*). Using mass spectrometry (MALDI-TOF/TOF), we confirmed that OVA-

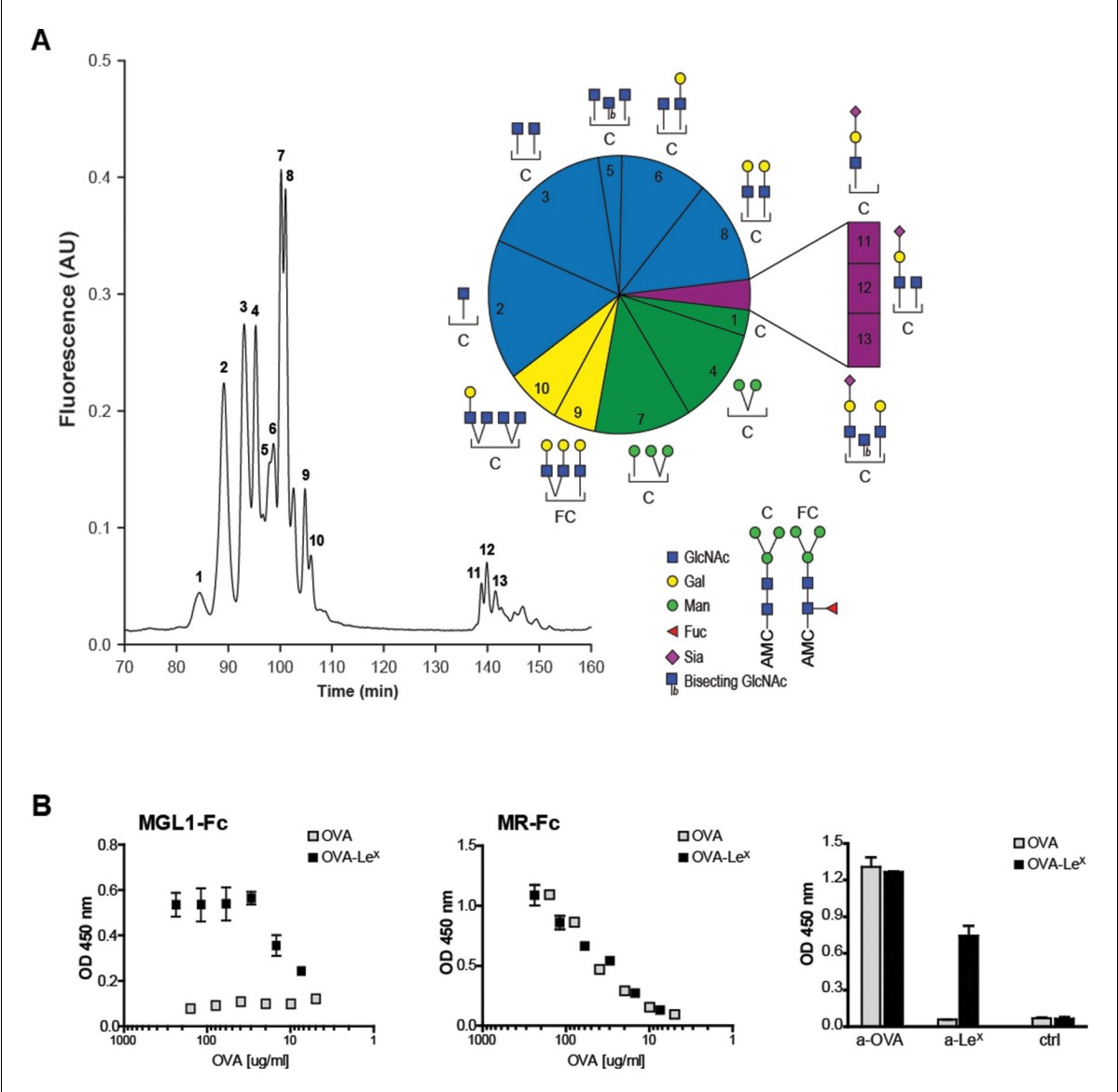

**Figure 1.** Generation of OVA-neo-glycoconjugates with Le$^X$ that confers binding of OVA to MGL1. (**A**) A glycan profile of OVA was generated using a multidimensional normal phase nano-HPLC coupled with an electrospray ionization interface mass spectrometer with an intercalated nanofluorescence detector. The different glycan species, indicated by numbers, are shown on the right; their relative proportion is represented in a pie chart. (**B**) ELISA showing functional modification of OVA with Le$^X$ glycans, as detected with anti-Le$^X$ antibodies and resulting in binding of MGL1-Fc. Unconjugated OVA does not carry any ligands for MGL1. Modification of OVA with Le$^X$ did not alter the ability to bind to MR as illustrated by equal binding kinetics of MR-Fc to native OVA and OVA-Le$^X$. OVA and OVA-Le$^X$ preparations contain similar amounts of OVA as detected with anti-OVA antibodies.

The following figure supplement is available for figure 1:

**Figure supplement 1.** The MALDI-TOF/TOF mass spectrum of OVA-Le$^X$ (Red) shows an increase of 1,2 KDa compared to unconjugated OVA (Blue), corresponding to addition of two Le$^X$ molecules per OVA molecule.

Le$^X$ increases to 1.2 KDa in mass, indicating that at least 40% of the total OVA-Le$^X$ preparation contained the Le$^X$ (*Figure 1—figure supplement 1*). This increase corresponded to the addition of two Le$^X$ molecules per OVA molecule.

Furthermore, using anti-Le$^X$ antibodies the presence of Le$^X$ was detected on OVA-Le$^X$, whereas absent on native glycosylated OVA, indicating that the glycan was successfully conjugated to OVA (*Figure 1B*). While native OVA interacted with the MR (MR-Fc, *Figure 1B*) and as earlier described

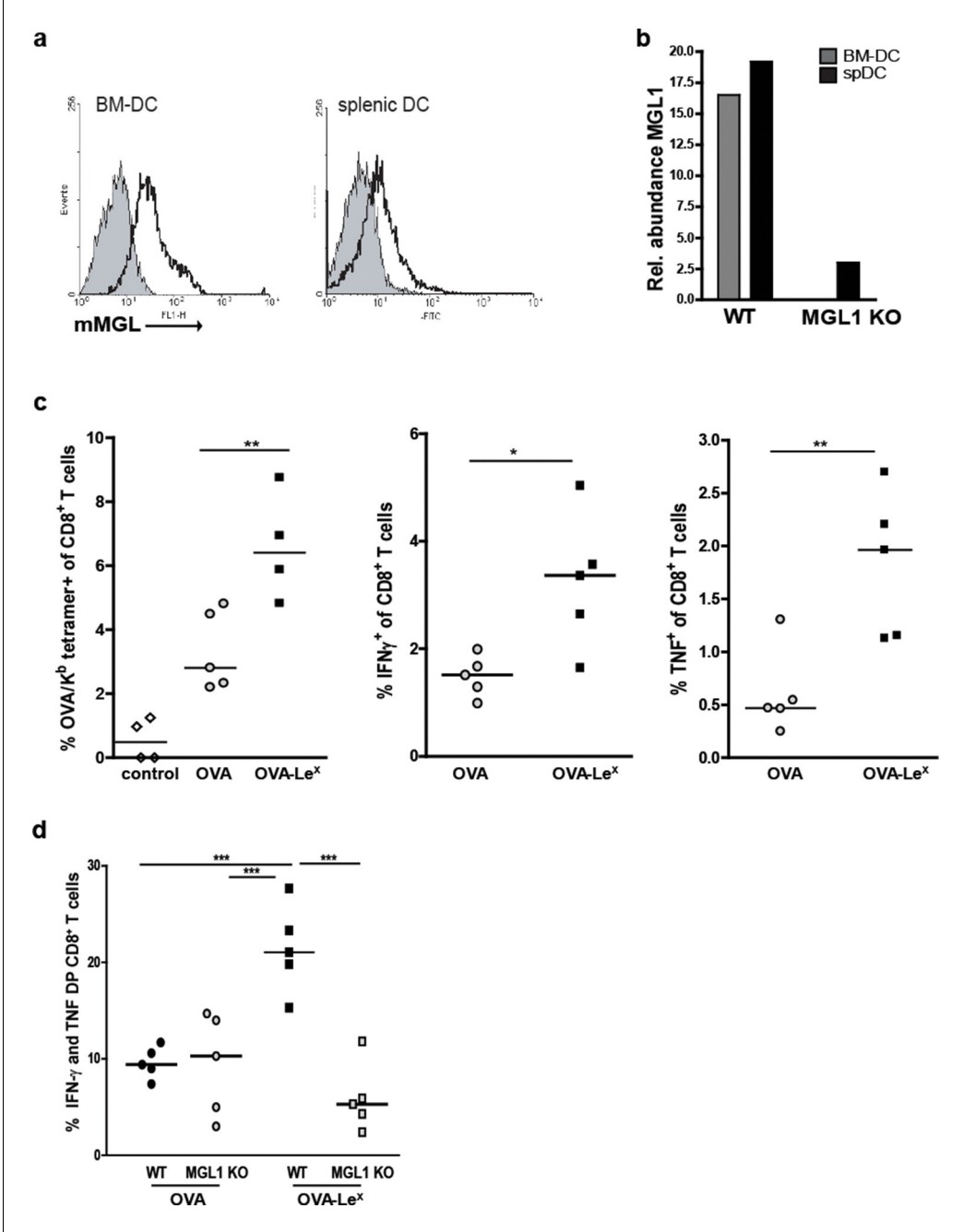

**Figure 2.** Immunization with OVA-Le[X] induces increased CD8[+] T cell responses in vivo. (**A**) Expression of murine MGL on BM-DCs and CD11c[+] spDCs was analyzed by flow cytometry. (**B**) MGL1 mRNA expression by BM-DC and splenic DC from WT and MGL1 KO mice was determined using qRT-PCR. GAPDH was used as a reference gene and the results are representative of three independent experiments. C57BL/6 mice were immunized *s.c.* with either OVA-Le[X] or native OVA mixed with anti-CD40 using a prime-boost protocol. Spleens were analyzed by flow cytometry to determine the frequency of (**C**) H2-K[b]/SIINFEKL-tetramer-binding CD8[+] T cells and IFN-γ or TNF production by activated CD8[+] T cells was determined by intracellular staining after OVA-specific re-stimulation ex vivo. Dots represent individual mice (n=4–5 mice/group; **p<0.01). Bars indicate median of each group. Graphs shown are representative of two independent experiments. (**D**) C57BL/6 and MGL1 KO mice were prime-boosted with either OVA-Le[X] or native OVA mixed with anti-CD40. Frequencies of IFN-γ and TNF-double-producing CD8[+] T cells were determined by intracellular staining after OVA-specific re-stimulation of splenocytes ex vivo. Dots represent individual mice (n=4–5 mice/group; *p<0.05 ***p<0.001). Bars indicate median of each group. Data are representative of 2 independent experiments.

*Figure 2 continued on next page*

*Figure 2 continued*

The following figure supplements are available for figure 2:

**Figure supplement 1.** Representative flow cytometry plots of (**A**) IFN-γ and (**B**) TNF- producing CD8$^+$ T cells in spleens of C57BL/6 mice that were immunized with either OVA-Le$^X$ or native OVA mixed with anti-CD40 using a prime-boost protocol; numbers above the gates designate the percentage of IFN-γ$^+$ or TNF$^+$ CD8$^+$ T cells.

**Figure supplement 2.** C57BL/6 and MGL1 KO mice were prime-boosted with either OVA-Le$^X$ or native OVA mixed with anti-CD40.

(*Burgdorf et al., 2007*), the conjugation of Le$^X$ to OVA conferred high-avidity binding to MGL1 as revealed using the soluble recombinant form of MGL1 (MGL1-Fc, *Figure 1B*). Moreover, the addition of Le$^X$ to OVA did not result in increased binding to the MR binding, indicating that Le$^X$ selectively binds MGL1. Similar binding with an anti-OVA antibody was determined, which indicates that similar protein concentrations in the preparations of OVA and OVA-Le$^X$ used in our study are present (*Figure 1B*, right panel). In conclusion, through glycan profiling we identified 23% mannose-rich glycans on native OVA that may facilitate MR binding. Addition of the glycan Le$^X$, next to these mannose glycans, on cysteine residues (two Le$^X$ moieties per molecule), conveyed MGL1 specificity to OVA.

## OVA-Le$^X$ binds MGL1 and augments priming of OVA-specific CD8$^+$ T cells in- vivo

We explored the potency of OVA-Le$^X$ to enhance T cell priming in vivo. Since DC are crucial for priming of naive T cells, we first established the presence of MGL on both splenic and bone marrow-derived DC (spDCs and BM-DCs) of C57BL/6 mice by staining with the anti-MGL antibody ER-MP23 (*Figure 2A*). Since the ER-MP23 antibody does not discriminate between the two murine MGL homologues MGL1 and MGL2, MGL1 expression in wild type (WT) CD11c$^+$ DCs was confirmed using qRT-PCR (*Figure 2B*). Immunization of mice with OVA-Le$^X$ mixed with agonistic anti-CD40 antibodies resulted in the enhanced priming of OVA-specific CD8$^+$ T cells as revealed from both higher numbers of OVA/H-2K$^b$ tetramer binding CD8$^+$ T cells and OVA-specific IFN-γ and TNF-producing CD8$^+$ T cells than obtained with native OVA (*Figure 2C*, *Figure 2—figure supplement 1*). Using MGL1KO mice, that lack MGL1 expression on DC (*Figure 2B*), we ascertained that MGL1 is the prime lectin involved in boosting the generation of antigen-specific effector T cells as upon immunization of these mice with Le$^X$-conjugated OVA no enhanced frequencies of OVA-specific IFN-γ and TNFα-double-producing T cells were detected (*Figure 2D*, *Figure 2—figure supplement 1*). In fact, the OVA-Le$^X$ immunized MGL1 KO mice displayed comparable numbers of effector T cells as WT mice immunized with native OVA. Together, these data show that the conjugation of two Le$^X$ glycans on OVA re-directs OVA to the CLR MGL1 and thereby enhances CD8$^+$ T cell priming in vivo.

## OVA-Le$^X$ induces Th1 skewing of naive CD4$^+$ T cells

Since we observed that Le$^X$-modified OVA increased priming of antigen-specific CD8$^+$ T cells we examined whether this also enhanced antigen-presentation to CD4$^+$ T cells. Both OVA-Le$^X$-loaded and native OVA-loaded spDCs induced CD4$^+$ OT-II T cell proliferation to a similar extent (*Figure 3A*), illustrating that the altered antigen uptake mediated by Le$^X$ did not affect loading on MHC class II molecules. Similar results were obtained using BM-DCs (*Figure 3A*). Although we did not observe any differential effect of Le$^X$ on CD4$^+$ T cell expansion, neoglycosylation of antigens could induce signaling via CLRs and herewith potentially influence Th cell differentiation (*Gringhuis et al., 2014*). We therefore investigated whether OVA-Le$^X$ affected the differentiation of naive CD4$^+$ T cells. Hereto BM-DCs and spDCs of C57BL/6 mice were pulsed with OVA-Le$^X$ and subsequently co-cultured with naive CD4$^+$CD62L$^{hi}$ OT-II cells. Co-cultures containing OVA-Le$^X$ loaded BM-DCs or spDCs contained significantly more IFN-γ-producing T cells than those containing OVA-loaded DCs (*Figure 3B*). Neither induction of IL-4- nor IL-17A-producing CD4$^+$ T cells was observed (*Figure 3B*, upper and middle panel and data not shown). In addition, induction of Foxp3$^+$ T cells was not detected (data not shown). To exclude that the Th1 skewing by OVA-Le$^X$ loaded DCs was

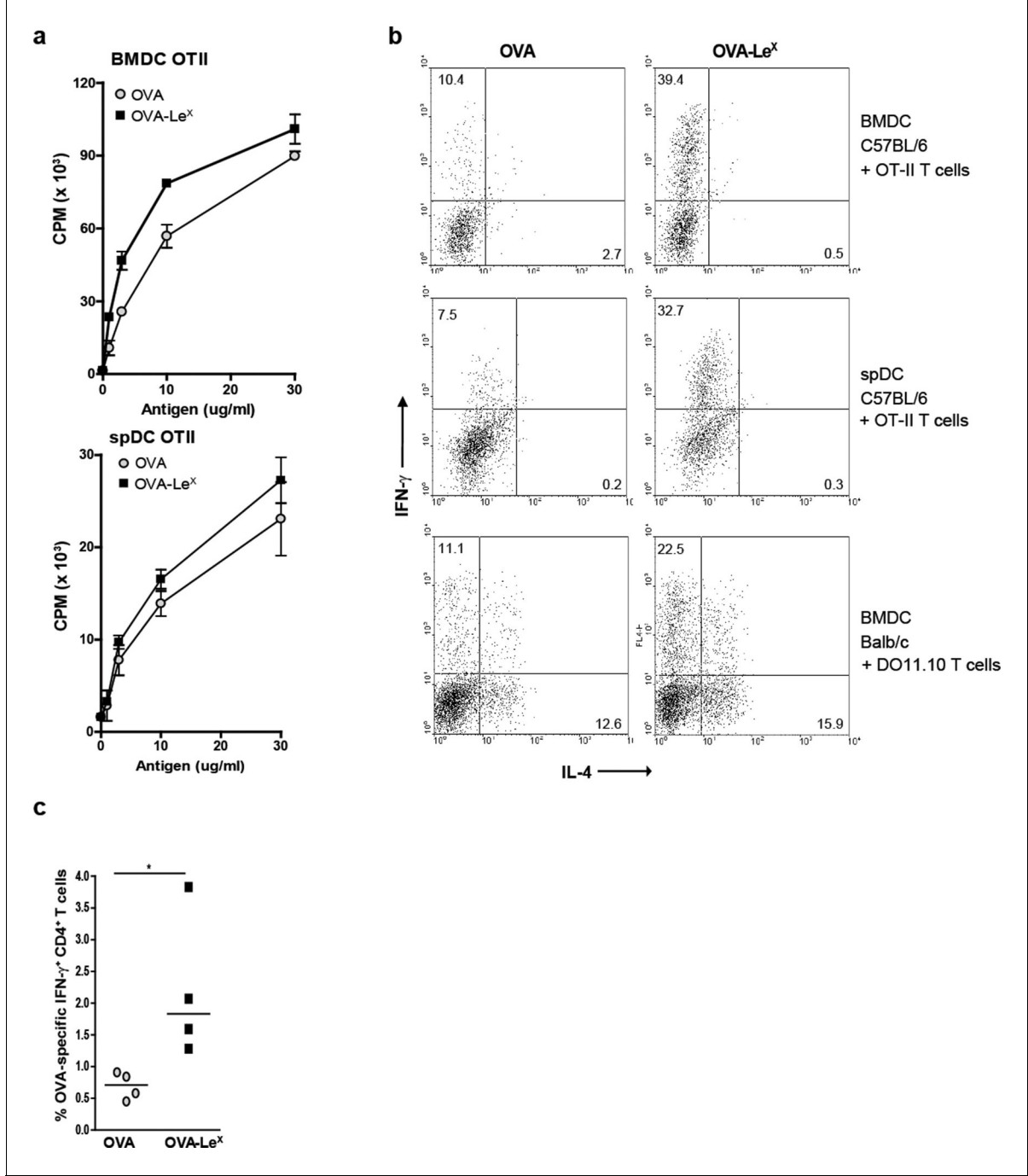

**Figure 3.** Modification of OVA with Le$^X$ structures skews naive CD4$^+$ T cells towards the Th1-effector lineage. (**A**) Pulsing of CD11c$^+$ spDCs or BM-DCs with OVA-Le$^X$ results in equal OT-II proliferation as native OVA. Expansion of OVA-specific T cells was determined using $^3$H-thymidine incorporation. Data are shown as mean ± SD of triplicate cultures, representative of three independent experiments. (**B**) Flow cytometric analysis of OT-II or DO11.10 T cells differentiated by OVA-Le$^X$ or OVA-loaded spDCs or BM-DCs. Cells were gated on CD4$^+$ T cells. Numbers in dot plots indicate the percentage of IFN-γ$^+$ or IL-4$^+$ of CD4$^+$ T cells. Dot plots are representative of five independent experiments. (**C**) C57BL/6 mice were immunized s.c. with either OVA-Le$^X$ or native OVA mixed with anti-CD40 using a prime-boost protocol and the frequency of IFN-γ-producing activated CD4$^+$ T cells in spleen was determined by intracellular staining after OVA-specific re-stimulation ex vivo. Dots represent individual mice, bars indicate median of each group (n=5 mice/group, **p<0.01). Graphs shown are representative of two independent experiments.

The following figure supplements are available for figure 3:

**Figure supplement 1.** No enhanced expansion of OT-II T cells when co-cultured for six days with OVA-Le$^X$ pulse-loaded DCs.

*Figure 3 continued on next page*

*Figure 3 continued*

**Figure supplement 2.** C57BL/6 mice were immunized s.c. with either OVA-Le[X] or native OVA mixed with anti-CD40 using a prime-boost protocol and the frequency of IFN-γ-producing activated CD4[+] T cells in spleen was determined by intracellular staining after OVA-specific re-stimulation ex vivo.

attributed to the more Th1 prone status of C57BL/6 (*Gervais et al., 1984*), we also performed the Th-differentiation assay with cells derived from Th2 prone BALB/c mice (*Hsieh et al., 1995*). We observed that naive OVA-specific CD4[+] T cells from DO11.10 Tg mice that were stimulated with OVA-loaded BM-DCs differentiated into IL-4 secreting T cells (*Figure 3B*, lower panels). However, the generation of IL-4-producing T cells was not influenced by loading DCs with OVA-Le[X] as these cultures contained comparable percentages of IL-4-producing DO11.10 T cells. Using these Th2-prone T cells, OVA-Le[X]-pulsed DCs still induced considerably more IFN-γ-producing CD4[+] T cells than native OVA-pulsed DCs (*Figure 3B*, lower panel). Since this assay takes three days longer than the antigen-presentation assay, it is possible that the higher frequency of IFN-γ-producing CD4[+] T cells is due to increased division of OVA-specific CD4[+] T cells. However we found that the amount of proliferation of OVA-specific CD4[+] T cells induced by stimulation with OVA-Le[X]-loaded DCs after 6 days is similar to that induced by OVA-loaded DCs (*Figure 3—figure supplement 1*). The augmented induction of CD4[+] Th1 cells was also observed in vivo as revealed from the higher frequencies of IFN-γ-producing OVA-specific CD4[+] T cells in the spleens of OVA-Le[X] immunized mice than in mice immunized with native OVA (*Figure 3C*, *Figure 3—figure supplement 2*). These data indicate that the increased numbers of Th1 cells induced by OVA-Le[X]-loaded DC are not due to increased proliferation of OT-II cells, but probably due to MGL1-mediated signaling.

## MGL1 mediates cross-presentation of OVA-Le[X] independently of TLR signaling

Since we observed a great enhancement of antigen-specific CD8[+] T cell priming in vivo when immunizing with OVA-Le[X] to target MGL1, we wanted to reveal the mechanism that regulates this augmented cross-priming. The internalization of OVA by BM-DC was significantly increased when OVA was modified with Le[X] (*Figure 4A—figure supplement 1*). To investigate whether the addition of Le[X] glycans affected the efficiency of cross-presentation of OVA by DC, we loaded murine BM-DC or spDC with OVA-Le[X] and measured their potency to present OVA-derived peptides in MHC class I by measuring the proliferation of OVA-specific OT-I T cells. Strikingly, both BM-DCs as well as spDCs induced substantially more proliferation of OT-I T cells when they were pulsed with OVA-Le[X] compared to native OVA (*Figure 4B—figure supplement 2*). Even at low concentrations of antigen (*i.e.* 7.5 μg/ml), the proliferating OT-I T cells were doubled (26.7% to 51.7%) using OVA-Le[X] compared to using OVA (*Figure 4B*), indicating that the modification of OVA with Le[X] greatly affected the cross-presentation of OVA. Moreover, detection of SIINFEKL/H-2K[b] complexes on the cell membrane of OVA-Le[X]-loaded DCs by staining with the 25.1D1 antibody confirmed enhanced antigen loading on MHC-I molecules and transportation to the cell-surface of internalized OVA-Le[X] compared to native OVA (*Figure 4C* + Figure supplement 3). Cross-presentation of OVA-Le[X] was clearly mediated by MGL1 as demonstrated using MGL1 KO BM-DCs or steady-state spDCs (*Figure 4D*).

Cross-presentation of OVA via the MR was shown previously to be dependent on TLR signaling and the presence of high amounts of antigen ([*Burgdorf et al., 2006*; *Burgdorf et al., 2008*; *Blander and Medzhitov, 2006*] and *Figure 4—figure supplement 4*, left panel). The observed differences in cross-presentation between OVA and OVA-Le[X] were not due to any potential contamination with the TLR4 ligand LPS, as both protein preparations did not trigger IL-8 production by TLR4-transfected HEK293 cells (*Figure 4—figure supplement 5*). In addition, both OVA preparations neither induced maturation of BM-DCs nor altered their cytokine production (data not shown). To exclude any potential role of TLR signaling on the MGL1-mediated cross-presentation of OVA-Le[X], we made use of BM-DCs from mice that lack both MyD88 and TRIF (*i.e.* MyD88/TRIF DKO). However, MyD88/TRIF DKO BM-DCs still induced more OT-I proliferation when targeted with OVA-Le[X] than with OVA (*Figure 4E*) and only a slight reduction of cross-presentation was observed compared to that induced by WT BM-DCs, suggesting a minor role for MyD88- or TRIF-signaling in MGL1-induced cross-presentation. In line with previous findings, neither exogenous loading of

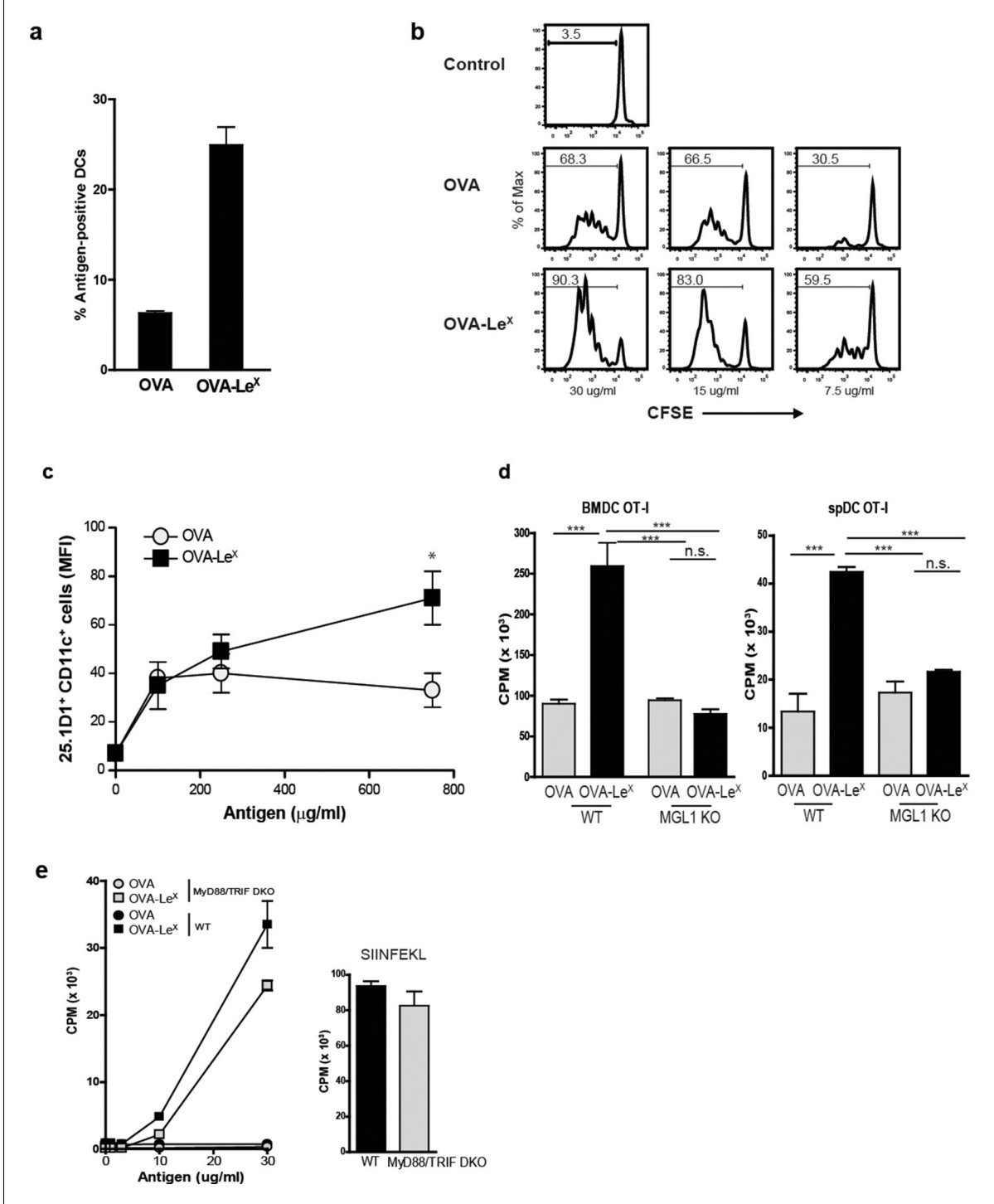

**Figure 4.** MGL1 mediates cross-presentation of OVA-Le[X] independently of TLR signaling. (A) Uptake of fluorescent-labeled OVA-Le[X] or native OVA (30 µg/ml) by WT BM-DCs was analyzed by flow cytometry after 90 min. Graphs indicate the mean ± SD of triplicates and are representative of three independent experiments. (B) CFSE-labeled OT-I T cells were incubated with BM-DCs pulse-loaded with indicated concentrations of OVA-Le[X] or OVA for 4h. Un-loaded DC served as controls. Proliferation of OT-I T cells was analyzed after 3 days by flow cytometry. Percentages of divided OT-I cells are indicated. (C) OVA-Le[X] induces more OVA257-264/H2-K[b] I complexes at the cell-surface of DCs than native OVA, as shown by 25.1D1 staining 18h after pulse loading of BM-DCs with OVA-Le[X] or native OVA. *p<0.05. (D) WT or MGL1 KO BM-DCs or CD11c[+] spDCs are pulsed with OVA-Le[X] (black bars) or native OVA and OT-I proliferation was determined on day 3 by [3H]-thymidine uptake. Data are presented as mean ± SD of triplicates, representative of three independent experiments. ***p<0.001, ns not significant. (E) Cross-presentation of OVA-Le[X] is independent of MyD88 and/or TRIF signaling. BM-DC from WT or MyD88/TRIF DKO mice were pulsed with indicated concentrations of antigen and co-cultured with OT-I T cells. DCs pulsed with
*Figure 4 continued on next page*

*Figure 4 continued*

the nominal epitope SIINFEKL served as controls (right panel). Proliferation was determined by [³H]-thymidine uptake. Data are representative of two experiments and indicated as mean ± SD of triplicates.

The following figure supplements are available for figure 4:

**Figure supplement 1.** Uptake of fluorescent-labeled OVA-Le$^X$ or native OVA (30 μg/ml) by WT BM-DCs was analyzed by flow cytometry after 90 min (upper panel).

**Figure supplement 2.** Enhanced cross-presentation of OVA-Le$^X$ by DCs as measured by ³H-Thymidine incorporation.

**Figure supplement 3.** Representative flow cytometry plots of 25.1D1 staining of BM-DCs 18h after pulse loading with OVA-Le$^X$ or native OVA (750 ug/ml).

**Figure supplement 4.** Cross-presentation of OVA requires TLR4 triggering and is TAP-dependent.

**Figure supplement 5.** OVA-Le$^X$ formulations are free of endotoxins.

MHC-I molecules with OVA257-264 peptides (*Figure 4E*) nor MHC class II presentation of OVA-Le$^X$ and OVA was dependent on MyD88- or TRIF- signaling and resulted in comparable expansion of OVA-specific T cells (data not shown).

## Cross-presentation induced by MGL1-targeting is independent of TAP-transport and Cathepsin-S -induced endosomal degradation

Several cross-presentation pathways have been described, one of which is dependent on the transport of peptides from the cytosol into MHC-class I loading compartments via TAP-molecules (*Amigorena and Savina, 2010*; *Adiko et al., 2015*), whereas another cross-presentation pathway depends on endosomal degradation by the cysteine protease Cathepsin-S (*Shen et al., 2004*). To study a role for TAP transporters in our model, BM-DCs of TAP1 KO and WT control mice were pulsed with OVA-Le$^X$ followed by incubation with OT-I T cells. Surprisingly, cross-presentation induced by OVA-Le$^X$ was not reduced by the absence of TAP as OT-I proliferation induced by OVA-Le$^X$-loaded TAP1 KO BM-DCs was not decreased compared to OVA-Le$^X$-loaded WT BM-DC (*Figure 5A*). In accordance with previous publications (*Burgdorf et al., 2008*), we showed that the administration of OVA with LPS is cross-presented in a TAP-dependent manner (*Figure 4—figure supplement 2*). Furthermore, the possibility that the results are confounded by reduced levels of MHC-class I on TAP1 KOBM-DCs were excluded as the presentation of exogenously loaded OVA257-264 peptide is equal by both WT and TAP1 KOBM-DCs (*Figure 5A*). In addition, we excluded the involvement of the Cathepsin-S pathway for cross-presentation of OVA-Le$^X$ as cross-presentation of OVA-Le$^X$ by BM-DCs from Cathepsin-S KO mice (Cat-S KO) was not reduced compared to WT BM-DCs (*Figure 5B*). As expected, the MHC-class II-restricted CD4$^+$ T cell proliferation was compromised in the Cat-S KO BM-DCs (data not shown), illustrating the involvement of Cathepsin-S in cleaving the invariant chain of the MHC-class II molecule (*Nakagawa et al., 1999*).

## Modification of OVA with LeX alters the intracellular routing of OVA

As the dominant cross-presentation of Le$^X$-modified OVA was neither dependent on TAP nor required TLR signaling, we hypothesized that this may be due to the altered uptake and intracellular routing of OVA in DCs. We therefore used imaging flow cytometry, a method that allows high-throughput image analysis of cells in flow with near-confocal resolution to analyze the intracellular routing of fluorescent labeled OVA-Le$^X$. Co-staining with markers for early endosomal (EEA-1), late endosomal/lysosomal (LAMP1) and recycling endosomal (Rab11) compartments illustrated a swift co-localization of OVA-Le$^X$ with EEA1 and Rab11 as shown by high co-localization scores at 15 min (*Figure 6A*, *Figure 6—figure supplement 1*). However, this co-localization score was strongly decreased at 60 min. At this time-point, higher co-localization scores were detected for OVA-Le$^X$ with LAMP1 and Rab11 (*Figure 6B*, *Figure 6—figure supplement 2*). We then further dissected the intracellular pathway of OVA-Le$^X$ using confocal laser scan microscopy (CLSM) and compared it to

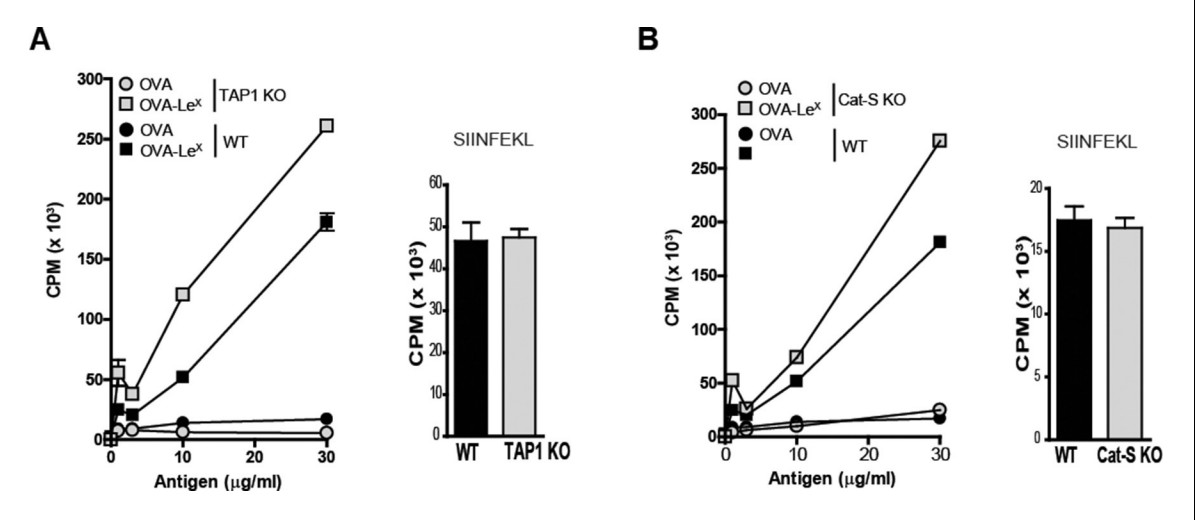

**Figure 5.** Le^X-modified antigen is cross-presented in a TAP- and Cathepsin-S-independent fashion. To examine whether cross-presentation of OVA-Le^X involves TAP or Cathepsin-S (**A**) TAP1 KO and (**B**) Cat-S KO BM-DCs and WT BM-DCs were pulsed with OVA-Le^X or native OVA and co-cultured with OT-I T cells for 3 days. DCs exogenously loaded with SIINFEKL for 3h served as control. Proliferation was determined by [3H]-Thymidine uptake and data are presented as mean ± SD of triplicates (representative of three experiments).

the intracellular routing of native OVA. We confirmed that native OVA, which internalized via MR, was routed to EEA1^+Rab11^+ compartments within 2h (*Figure 6C*, upper-left panel and [*Burgdorf et al., 2008*]). However, the neoglycosylation with Le^X altered the intracellular routing of OVA, showing its presence predominantly within LAMP1^+Rab11^+ compartments (*Figure 6C*, upper-right panel). Moreover, OVA-Le^X internalized by BM-DC from MGL1 KO mice was routed to EEA1^+-Rab11^+ compartments and did not end up in LAMP1^+Rab11^+ compartments, similar to OVA in WT BM-DC (*Figure 6C*, lower panels + *Figure 6—figure supplement 3*). Together, these data suggest that upon internalization, OVA-Le^X is rapidly shuttled to Rab11^+EEA1^+ compartments from where it moves to Rab11^+LAMP1^+ compartments, where it persists for longer periods (*i.e.* >24h). Thus, these results indicate that internalization via MGL1 allows the antigen to enter the endosomal/lysosomal pathway.

The development of early endosome into late endosome/lysosome coincides with a decreasing pH gradient. The pH at which MGL1 dissociates from its ligand is indicative of the compartment in which the antigen becomes available for degradation and loading on MHC-molecules. We therefore analyzed MGL1-binding to its ligand Le^X at different pH that resembled the pH of the intracellular compartments. MGL1 starts dissociating from Le^X already at pH 6.5 (*Figure 6D*). This indicates that OVA-Le^X becomes available for degradation in the early and late endosomal compartments, compartments both associated with cross-presentation.

The sustained presence of OVA-Le^X in Rab11^+LAMP1^+ compartments (*Figure 6B+C*) prompted us to investigate whether these compartments facilitate prolonged cross-presentation, as shown by van Montfoort *et al.* for OVA-immune complexes (*van Montfoort et al., 2009*). Indeed, even two days after antigen pulse, OVA-Le^X-loaded DC induced strong expansion of OT-I T cells, suggesting that OVA-Le^X-loaded DC have prolonged cross-presentation capacity (*Figure 7A*). At the highest antigen concentration used (*i.e.* 30 µg/ml), the percentage of proliferated OT-I T cells was only slightly reduced 48 h after pulse compared to that induced by DCs that were pulsed with OVA-Le^X for 4 h (*Figure 7A*). Expansion of OT-I T cells driven by 48 h pulse loaded DCs was even detectable at low antigen concentrations (3.75 µg/ml). OT-II proliferation induced by DCs pulsed with OVA-Le^X for 48h was also still detectable although less pronounced than the OT-I induced proliferation (*Figure 7B*), suggesting that the prolonged storage of antigen in these intracellular compartments predominantly favored cross-presentation.

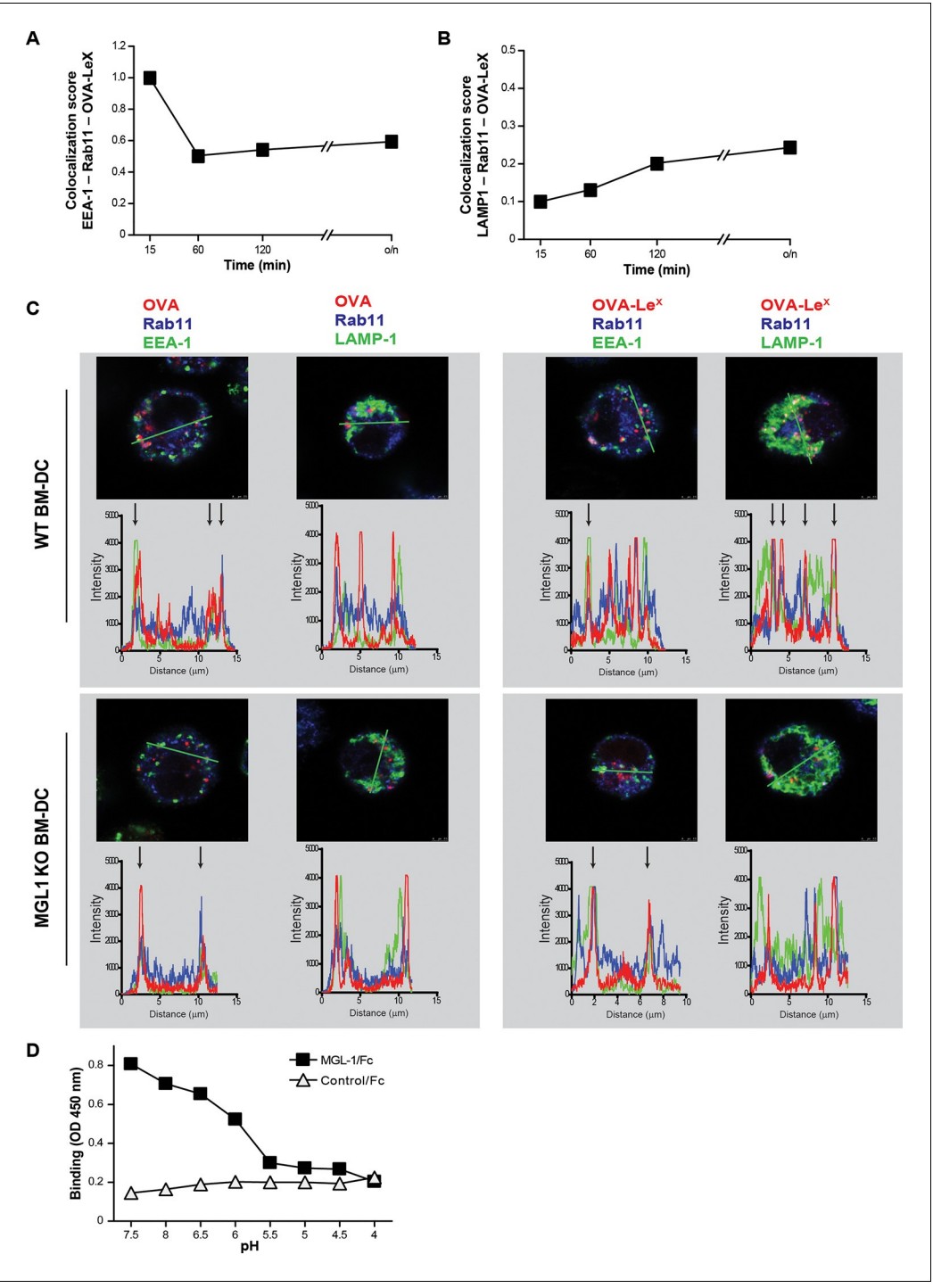

**Figure 6.** OVA-Le^X is routed to Rab11+LAMP1+ compartments where it is stored for presentation in MHC class I. (**A, B**) WT BM-DCs were pulsed with AlexaFluor 674-OVA-Le^X (30 μg/ml) and chased at the indicated time-points to assess triple co-localization scores of OVA-Le^X with (**A**) EEA-1 and Rab11 or (**B**) LAMP1 and Rab11 using imaging flow cytometry. (**C**) WT (upper panels) and MGL1 KO (lower panels) BM-DCs were incubated with Dylight-633-OVA-Le^X or native OVA (30 μg/ml) and 2h later co-localization of OVA antigen (Red) with early endosomal (EEA-1, Green) or endosomal/lysosomal (LAMP1, Green) and recycling endosomal (Rab11, Blue) compartments was analyzed using CSLM. From a z-stack, histograms were created for a selected area (indicated by a line, upper part of each panel) using the Leica confocal software. Histograms were created from each fluorochrome and overlays were made by the program. Arrows indicate co-localization of antigen (Red) with EEA1&Rab11 or LAMP1&Rab11. (**D**) MGL1-Fc binding to Le^X-PAA was determined at indicated pH by ELISA.

*Figure 6 continued on next page*

*Figure 6 continued*

The following figure supplements are available for figure 6:

**Figure supplement 1.** Examples of WT BM-DCs pulsed with AlexaFluor 674-OVA-Le^X displaying high co-localization of OVA-Le^X with EEA-1 and Rab11 as measured by imaging flow cytometry.

**Figure supplement 2.** Examples of WT BM-DCs pulsed with AlexaFluor 674-OVA-Le^X displaying high co-localization of OVA-Le^X with LAMP1 and Rab11 as measured by imaging flow cytometry.

**Figure supplement 3.** WT (upper panels) and MGL1 KO (lower panels) BM-DCs were incubated with Dylight-633-OVA-Le^X or native OVA (30 μg/ml) and 0.5 h and 2 h later co-localization of OVA antigen (Red) with early endosomal (EEA-1, Green) or endosomal/lysosomal (LAMP1, Green) and recycling endosomal (Rab11, Blue) compartments was analyzed using CSLM.

Taken together, these data suggest that antigen internalized by MGL1 is routed from RAb11$^+$-EEA1$^+$ compartments towards Rab11$^+$LAMP1$^+$ compartments, which seem to associate with the extended antigen-processing and cross-presentation.

## Discussion

We here demonstrate that the glycosylation-profile of antigens has a major influence on antigen uptake and intracellular compartmentalization, thereby affecting both antigen presentation and the type and strength of the induced immune response. Modification of the model-antigen OVA with Le^X glycans directs OVA towards MGL1, skewing naive CD4$^+$ T cell differentiation towards IFNγ-producing Th1 cells. Moreover, targeting OVA to MGL1 through the conjugation of Le^X, substantially enhanced cross-presentation as revealed by the increased frequency of OVA-specific CD8$^+$ effector T cells in vitro and in vivo. Importantly, MGL1-dependent cross-presentation occurred at low antigen dose and independently of TLR-signaling. Moreover, this cross-presentation pathway did not involve TAP-transporters and Cathepsin-S. MGL1 targeting involved antigen routing to a Rab11$^+$LAMP1$^+$ compartment in which antigen was present for prolonged periods.

Previous studies on CLR-mediated antigen uptake and cross-presentation, in particular by the MR, demonstrated a clear requirement for a TLR ligand (*Kovacsovics-Bankowski and Rock, 1995*; *Burgdorf et al., 2008*; *Sancho et al., 2008*; *Segura et al., 2009*). A common denominator in some of these studies is that the use of antibody–antigen conjugates could potentially induce a different signal than the natural ligand, due to binding to different part of the receptor or through co-engagement of Fc-receptors. The fact that the addition of Le^X glycans to OVA obviates the need for TLR signals for the induction of cross-presentation and Th1 priming in vitro, may indicate that MGL1 signaling is involved in these processes. Some CLR, like DC-SIGN, Clec9A and Dectin-1 are known to induce signaling after triggering by their natural ligands (*Geijtenbeek and Gringhuis, 2009*; *Sancho et al., 2009*). Till now, no signaling pathway has been described for MGL1. Triggering of huMGL resulted in ERK1/2 and NF-κB activation, and results in elevated levels of IL-10 and TNF (*Li et al., 2012*; *van Vliet et al., 2013*). In our studies, the uptake of Le^X-modified OVA through MGL1 was not associated with any DC maturation or altered cytokine production by DCs. The fact that OVA-Le^X induced an enhanced frequency of Th1 cells in vitro and of antigen-specific effector T cells in vivo when combined with agonistic anti-CD40 Abs illustrates that a yet to be determined costimulatory signal is essential for the induction of effector CD8$^+$ and CD4$^+$ T cells. This powerful function of MGL1 to establish antigen-specific immunity, stands opposite to its recently demonstrated anti-inflammatory function, which include induction of IL-10 production and altered adhesive function by APC (*Saba et al., 2009*; *Westcott et al., 2009*).

Various models for cross-presentation of antigens have been contemplated. The 'cytosolic pathway' of cross-presentation allows receptor-mediated endocytosis or phagocytosis and antigen translocation into the cytosol, where they are degraded into antigenic peptides by the proteasome and transported into the lumen of the endoplasmic reticulum (ER) (*Kovacsovics-Bankowski and Rock, 1995*; *Shen et al., 2004*) or ER/phagosomal fusion compartments (*Guermonprez et al.,*

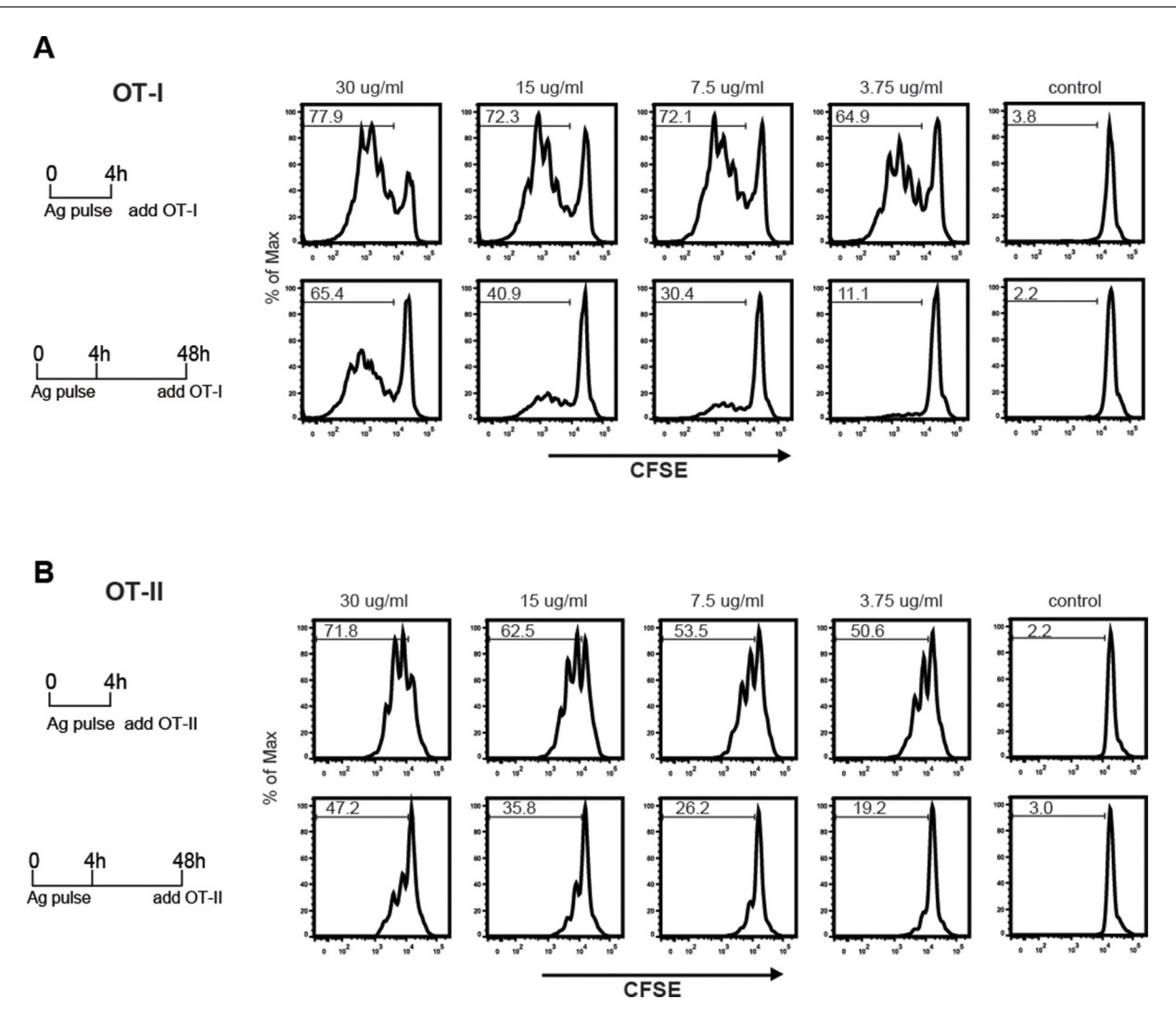

**Figure 7.** MGL1 targeting with OVA-Le$^X$ shows sustained antigen presentation in MHC-I. WT BM-DCs were pulsed for 4 h with titrated amounts of OVA-Le$^X$ and washed with culture medium. DCs were then chased for 48 h in antigen-free medium. (**A**) BM-DCs pulsed for 4h with OVA-Le$^X$ induced MHC-I antigen presentation as measured by CFSE-labeled OVA-specific OT-I cells (upper panel). Sustained presentation is shown after 48 h (lower panel). (**B**) MHC-II antigen presentation 4h and 48h after pulse-loading with OVA-Le$^X$, analyzed by OT-II proliferation. Data are presented as percentage of proliferated T cells and representative of three independent experiments.

2003; *Houde et al., 2003*) by TAP-transporters. A variation to the cytosolic pathway was described in which soluble OVA is taken up via the MR and is supplied to stable early endosomal compartments (*Burgdorf et al., 2007*). In these compartments, the MHC class I machinery as well as TAP-transporters are selectively recruited, facilitating direct loading of antigen-derived peptides onto MHC class I molecules without trafficking to the ER. The 'vacuolar pathway' allows antigens to be degraded into peptides in early endosomal compartments by endosomal proteases like Cathepsin-S, to be loaded on MHC class I molecules in the same compartment, thus is TAP-independent (*Shen et al., 2004*). The current described models for cross-presentation depend on peptide-transport by TAP from the cytosol to either ER or ER/phagosomal fusion compartments or on endosomal degradation of antigens via Cathepsin-S (reviewed in [*Amigorena and Savina, 2010*; *Burgdorf and Kurts, 2008*; *Mintern et al., 2015*; *Adiko et al., 2015*]). Although we observed localization of OVA-Le$^X$ in early endosomes, absence of either TAP molecules or Cathepsin-S did not influence the enhanced cross-presentation of OVA-Le$^X$. Our data indicate that the MGL1 cross-presentation

pathway is different than that of the MR and may occur via antigen storage compartments similar to those described for FcγR-targeted antigens (*van Montfoort et al., 2009*). We observed prolonged presence in Rab11[+]LAMP1[+] compartments in the absence of a maturation stimulus or apparent DC maturation. It is tempting to speculate that in these compartments cross-presentation of OVA is facilitated by Rab11 as it was recently shown that Rab11 activity mediates delivery of MHC-I molecules to phagosomes used for cross-presentation during infection (*Nair-Gupta et al., 2014*). However, recruitment of additional components involved in antigen cross-presentation by the SNARE Sec22b40 complex cannot be excluded (*Cebrian et al., 2011*). The prolonged antigen cross-presentation via MGL1 allows sustained cross-priming of CD8[+] T cells in vivo and thus the efficacy of tumor vaccines that target MGL1 may be higher than of conventional vaccines containing non-targeted antigens.

In the current study, we show that already at very low concentrations, soluble protein antigens are efficiently cross-presented upon uptake by MGL1. DCs pulsed with a low dose of OVA-Le[X] (7.5 µg/ml) induced significant CD8[+] T cell proliferation, whereas the same concentration of native OVA was hardly cross-presented (*Figure 4*). Others have demonstrated that native OVA can only be cross-presented when a high dose was used (0.5–1.0 mg/ml) and when accompanied with TLR-triggering (*Burgdorf et al., 2008*; *Segura et al., 2009*). Although the glycan-modification was associated with increased antigen uptake and enhanced cross-presentation, it did not result in enhanced presentation of antigen on MHC-class II indicating that increased uptake of antigen is not per se a requirement to facilitate MHC class I and II presentation. Therefore, it is most likely that the CLR (and thus the glycosylation of the antigen) dictates how efficiently an antigen is cross-presented. Based on our study, MGL1 cross-presents antigen much more efficiently than the MR.

The activity of the glycosylation machinery is subjected to subtle regulation and depends on the cell-type or activation status of the cell. Upon malignant transformation glycan profiles may change dramatically. Le[X] carbohydrate structures are described to be expressed in the brain (*Lieberoth et al., 2009*), and can be *de novo* expressed on cancer stem cells (*Read et al., 2009*) or pathogens, and the glycosylation pattern of in vivo accumulating antigens, being brain tissue, tumor tissue or pathogen structure, is crucial for directing specific CLR antigen uptake and cross-presentation (*Hittelet et al., 2003*; *Ohana-Malka et al., 2003*). Although in our studies only two Le[X] glycans were conjugated to each OVA molecule, it cannot be ruled out that multivalent presentation of Le[X], such as often observed on tumors or pathogens (*Bergman et al., 2004*; *Saeland and van, 2011*), may alter avidity-induced MGL-1 signaling and antigen presentation, and induce an anti-inflammatory immune repertoire. At this stage, we do not know whether high glycan valency further enhances or inhibits the MGL1-mediated (cross-)priming.

The fact that pulsing of DCs with OVA-Le[X] resulted in improved cross-priming and Th1-skewing indicates that conjugation of these carbohydrates to tumor-antigens could be beneficial for the induction of potent anti-tumor responses. Therefore, the use of glycans for targeting CLR for anti-cancer immunotherapy could have several advantages. First, they mimic natural function of the receptors, inducing 'natural' signaling cascades in DCs. Furthermore, they are very small molecules, easy to use and relatively cheap to produce. Importantly, many glycans may be considered self-antigens, in contrast to recombinant antibodies that are often not completely of human origin. These properties make it possible to decorate any antigen of choice with glycans to target specific receptors.

In conclusion, our studies indicate that the glycan composition of protein antigens is of fundamental importance in dictating the intracellular routing and Th-skewing, and should be considered as a major determinant in the design of therapeutic vaccines against cancer and infectious diseases.

## Materials and methods

### Mice

C57BL/6 mice (Charles River Laboratories) were used at 8–12 weeks of age. MGL1KO mice, which have a null mutation within the *Clec10a* gene, are on the C56BL/6 background and were kindly provided through the Consortium for Functional Glycomics. TAP1 KO mice have a null mutation within the *Tap1* gene. MGL1 KO, TAP1 KO, OT-I and OT-II TCR transgenic mice were bred in our animal

facilities under specific pathogen-free conditions. All experiments were performed according to institutional, state and federal guidelines.

## Antibodies and Fc-chimeric constructs

Fluorochrome-conjugated antibodies used: anti-CD11c-APC, anti-IFNγ-APC, anti-TNFα-PE-cy7, anti-IL-4-PE, anti-IL17-PE, anti-Foxp3-PE (e-Bioscience, Vienna, Austria), and anti-LAMP-1-v450 (BD-Pharmingen). Unconjugated mouse anti-OVA (Sigma Aldrich), mouse anti-Le$^X$ (Calbiochem), rat anti-mMGL (ER-MP23; kind gift from Dr. P. Leenen, Erasmus MC, Rotterdam, The Netherlands), rat anti-LAMP1 (BD-Pharmingen), rabbit anti-Rab11 (Life Technologies), goat anti-EEA1 (Santa Cruz Biotechnology) and rabbit anti-EEA-1 (Dianova). Secondary antibodies used: peroxidase-labeled F(ab')$_2$ fragment goat anti-human IgG, F(ab')$_2$ fragment goat anti-mouse IgG, (Jackson), peroxidase-labeled goat anti-mouse IgM (Nordic Immunology), goat anti-rat Alexa 448, goat anti-rat Alexa 647, donkey anti-goat Alexa 488, donkey anti-goat Alexa 647, donkey anti-rabbit Alexa 555 and donkey anti-rabbit Alexa 488 (Molecular Probes). MGL-1-Fc was generated as described earlier (*Singh et al., 2009b*). MR-Fc was kindly provided by L. Martinez-Pomares (University of Nottingham, Nottingham, UK).

## Generation of neo-glycoconjugates

Le$^X$ (lacto-N-fucopentose III; Dextra Labs, UK) carbohydrate structures were conjugated to OVA (Calbiochem) as previously described (*Singh et al., 2009a*). In short; the bifunctional cross-linker (4-*N*-maleimidophenyl) butyric acid hydrazide (MPBH; Pierce) was covalently linked to the reducing end of the Le$^X$ and the maleimide moiety of the linker was later used for coupling the Le$^X$ to OVA. Neo-glycoconjugates were separated from reaction by-products using PD-10 desalting columns (Pierce). Additionally, a Dylight 549-*N*-hydroxysuccimide (NHS) label (Thermo Scientific) was covalently coupled with OVA or OVA-Le$^X$ (Dylight-549-OVA). Free label was removed using a PD-10 column (Pierce).

The presence of Le$^X$ and CLR binding to OVA was measured by ELISA. In brief, OVA-conjugates were coated directly on ELISA plates (NUNC) and binding of MR-Fc, MGL1-Fc, anti-Le$^X$ and anti-OVA antibodies to OVA was determined as described (*Singh et al., 2009a*; *Hawiger et al., 2001*). The presence of endotoxin was measured using a LAL assay (Lonza) following manufacturer's protocol.

## Glycan analysis

OVA was deglycosylated by incubation in 5 IU of PNGase F (Roche Applied Sciences) o/n at 37°C. Proteins were extracted by reverse phase chromatography using Sep-Pak Vac C18 disposable cartridges (Waters). Glycans were further purified by reverse phase chromatography using Superclean ENVI-Carb cartridges disposable columns (Supelco). Glycans were lyophilized and re-dissolved in 30 ml of 7-Amino-4-methylcoumarin (160 mM, Sigma Aldrich) and 2-Picoline borane (270 mM, Sigma Aldrich) in DMSO:acetic acid (4:1, Riedel deHaën). 4-AMC-labelled glycans were purified by size exclusion chromatography using a Bio-Gel P2 (Bio-Rad) column with 50 mM ammonium formate (Sigma Aldrich) as running buffer. 4-AMC-labelled glycans were lyophilized and analyzed by multidimensional normal phase HPLC (UltiMate 3000 nanoLC, Dionex) using a Prevail Carbohydrate ES 0.075 x 200 mm column (Grace) coupled with an LCQ Deca XP with electrospray interface mass spectrometer (Thermo Finnigan) tuned with maltoheptoase (Sigma Aldrich) labeled with 4-AMC and with an intercalated fluorescence detector (Jasco FP-2020 Plus, Jasco) (maximum excitation 350 nm, band width 40 nm; maximum emission 448 nm, band width 40 nm) as previously described (*Kalay et al., 2012*).

## Molecular weight determination

Matrix-assisted laser desorption/ionization-time of flight (MALDI-TOF) mass spectrometry measurements were done using a 4800 MALDI-TOF/TOF Analyzer (Applied Biosystems). Mass spectra were recorded in the range from to 19,000 to 155,000 m/z in the linear positive ion mode. The data were recorded using 4000 Series Explorer Software and processed with Data Explorer Software version 4.9 (all from Applied Biosystems).

## Immunization of mice

C57BL/6 or Mgl1$^{-/-}$ mice were injected *s.c.* either with 100 µg OVA-Le$^X$ or OVA mixed with 25 µg anti-CD40 Ab (1C10) on day 0 and day 14. Mice were sacrificed one week after boost and the amount of antigen-specific CD8$^+$ T cells was analyzed in the spleen by staining with H2-Kb-SIINFEKL tetramers (Sanquin). Additionally, frequencies of OVA-specific cytokine-secreting T cells were analyzed by flow cytometry. Hereto, spleen cells were re-stimulated overnight with either 2 µg/ml SIIN-FEKL or 200 µg/ml EKLTEWTSSNMEER OVA peptides in the presence of 5 µg/ml Brefeldin A, then IFN-$\gamma$ and TNF$\alpha$ expression were assessed by intracellular staining using specific antibodies.

## Cells

BM-DC were cultured as previously described (*Singh et al., 2009a*). BM of *Myd88/Ticam1* DKO (referred to as MyD88/TRIF DKO) and *Ctss$^{-/-}$* (referred to as Cat-S KO) mice was kindly provided by Dr. T. Sparwasser (Twincore, Hannover, Germany) and Dr. K. Rock (Massachusetts Medical School, Worcester, MA, USA), respectively. CD11c$^+$ spDCs were isolated as previously described (*Singh et al., 2009a*). OVA-specific CD4$^+$ and CD8$^+$ T cells were isolated from spleen and lymph nodes cell suspensions from OT-II and OT-I mice, respectively, using the mouse CD4 and CD8 negative isolation kit (Invitrogen, CA, USA) according to manufacturer's protocol. T cell proliferation assays were performed as described (*Singh et al., 2009a*). In short, DCs were pulsed with OVA-Le$^X$ or OVA for 4h before incubation with OVA-specific OT-I or OT-II T cells (2:1 DC:T). [$^3$H]-Thymidine (1 µCi/well; Amersham Biosciences) was present during the last 16h of a 72h culture. [$^3$H]-Thymidine incorporation was measured using a Wallac microbeta counter (Perkin-Elmer). Alternatively, OT-I or OT-II T cells were labeled with CFSE and after 3 days dilution of CFSE was analyzed by flow cytometry. Differentiation of naive OT-II T cells, induced by OVA-Le$^X$ or OVA -pulsed BM-DCs or spDCs, was measured by an in vitro Th differentiation assay described earlier (*Singh et al., 2011*).

## cDNA synthesis and Real time PCR

mRNA was isolated by capturing poly(A$^+$)RNA in streptavidin-coated tubes using a mRNA Capture kit (Roche, Basel, Switzerland). cDNA was synthesized using the Reverse Transcription System kit (Promega, WI, USA) following the manufacturer's guidelines.

Real time PCR reactions were performed using the SYBR Green method in an ABI 7900HT sequence detection system (Applied Biosystems).

## Confocal Microscopy and imaging flow cytometry

BM-DCs were incubated with 30 µg/ml Dylight-633-OVA or OVA-Le$^X$ for 30min or 2h at 37°C, fixed and permeabilized for 20 min on ice, and stained with primary and secondary antibodies. Co-localization was analyzed using a confocal laser scanning microscope (Leica SP5 STED) system containing a 63x objective lens; images were acquired in 10x magnification and processed with Leica LAS AF software.

For imaging flow cytometry, approximately 1x10$^6$ BM-DCs were incubated with OVA-Le$^X$ for 30 min at 4°C, washed twice in ice-cold PBS and then transferred to 37°C. At the indicated time-points, cells were washed twice and fixated in ice-cold 4% paraformaldehyde (PFA, Electron Micros-copy Sciences) in PBS for 20 min. The cells were then permeabilized in 0.1% saponin (Sigma) in PBS for 30 min at RT and subsequently blocked using PBS containing 0.1% saponin and 2% BSA for 30 min at RT. Stainings were performed at room temperature (RT) in PBS supplemented with 0.1% saponin and 2% BSA. After staining, cells were washed twice in PBS, resuspended in PBS containing 1% BSA and 0.02% NaN3 and kept at 4°C until analysis. The cells were acquired on an ImageStream X100 (Amnis) imaging flow cytometer. A minimum of 15,000 cells was acquired per sample at a flow rate ranging between 50 and 100 cells/second at 60x magnification. At least 2000 cells were acquired from single-stained samples to allow for compensation. Analysis was performed using the IDEAS v6.1 software (Amnis). The cells were first gated based on the Gradient RMS (brightfield) feature and then based on area *vs* aspect ratio intensity (both on brightfield). The first gating identified the cells that appeared in focus, while the second excluded doublets and cells other than BM-DCs. 3-colour co-localization was calculated using the bright detail co-localization 3 feature.

## pH dependency of MGL1 binding

The pH dependency of MGL1 binding to Le$^X$ on antigens was determined by ELISA. Hereto, Le$^X$-PAA (Lectinity Holdings) was coated onto NUNC Maxisorp plates o/n at RT. The plates were blocked with 1% BSA in TSM buffer (20 mM Tris-HCl; 150 mM NaCl; 2 mM CaCl$_2$; 2 mM MgCl$_2$). After washing, MGL1-Fc was added in TSA with different pH and were kept in this buffer throughout the assay. Binding was detected using peroxidase-labeled F(ab')$_2$ fragment goat anti-human IgG.

## Statistical analysis

Graphpad prism 5.0 was used for statistical analysis. The Student's t-test and one-way ANOVA with Bonferroni correction were used to determine statistical significance. Statistical significance was defined as $p < 0.05$.

## Acknowledgements

We thank Sandra van Vliet for critical reading of the manuscript. This research was supported by VENI-NWO-ALW (grant 863.08.020 to JJGV), Senternovem SII071030 to WWJU, Mozaiek 017.001.136 to SKS, AICR 07-0163 to ES, ZonMw TOP (grant 91211011 to N.I.H)

## Additional information

### Funding

| Funder | Grant reference number | Author |
|---|---|---|
| ZonMw TOP | 91211011 | Nataschja I Ho |
| Senternovem | SII071030 | Manja Litjens<br>Wendy WJ Unger |
| NWO Mozaik grant | 017.001.136 | Satwinder Kaur Singh |
| American Institute for Cancer Research | 07-0163 | Eirikur Saeland |
| NWO VENI-ALW | 863.08.020 | Juan J Garcia-Vallejo |

The funders had no role in study design, data collection and interpretation, or the decision to submit the work for publication.

### Author contributions

IS-O, Designed and performed experiments, Acquisition of data, Analysis and interpretation of data, Drafting or revising the article; NIH, MAB, Performed experiments, Acquisition of data, Analysis and interpretation of data; ML, Performed experiments, Acquisition of data; HK, Produced glycan-antigen conjugates, Acquisition of data; LAMC, Acquisition of data, Analysis and interpretation of data; SKS, Performed experiments, Analysis and interpretation of data; ES, Designed experiments, Analysis and interpretation of data; JJG-V, Performed glycan-analysis and imaging flow cytometric analysis, Acquisition of data, Analysis and interpretation of data, Drafting or revising the article; FAO, YvK, Conception and design, Analysis and interpretation of data, Drafting or revising the article; WWJU, Conception and design, Analysis and interpretation of data, Drafting or revising the article, Supervised the study

### Author ORCIDs

Yvette van Kooyk, http://orcid.org/0000-0001-5997-3665

### Ethics

Animal experimentation: This study was performed in strict accordance with the recommendations in the Guide for the Care and Use of Laboratory Animals of the The Netherlands. All animal experiments were reviewed and approved by the Vrije University Scientific and Ethics Committees

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
