## [Decision Letter]

Thank you for submitting your work entitled "Glycan modification of antigen redirects its intracellular routing in dendritic cells, affecting priming of T cells" for consideration by *eLife*. Your article has been reviewed by three peer reviewers, and the evaluation has been overseen by Urszula Krzych as the Reviewing Editor and Tadatsugu Taniguchi as the Senior Editor.

The reviewers have discussed the reviews with one another and the Reviewing editor has drafted this decision to help you prepare a revised submission.

Summary:

This is an interesting and novel study that identifies a previously unknown pathway of antigen cross-presentation and demonstrates the role of specific glycan structures in targeting an antigen to this novel pathway. The use of the Lewis^x^ (Le^x^) modified OVA as an antigen, with the advantage of selective and specific Le^x^ addition to two know cysteine residues is a creative and informative approach. Le^x^ is a physiologically relevant glycosylation that is often found in infected cells and transformed cells. Specifically, the authors show that dendritic cells express MGL-1, the receptor for Le^x^ glycosylated molecules and that promotes an increased Ova-Le^x^ uptake and eventual processing that leads to CD8 T cell priming. The authors claim this is due to enhanced presentation of the relevant Kb/SIINFEKL complex following Ova-Le^x^ uptake compared to its native ovalbumin counterpart. The authors also claim that Le^x^ modification redirects routing of ovalbumin into the endolysosomal compartment where the antigen is available for MHC-I and -II presentation for at least 48h, the consequence of which is OVA-Le^x^ induced enhanced Th1 cell differentiation.

This work is timely and interesting for the immunological community because it extends on the finding that glycan receptor usage is an important parameter governing the extent of cross-presentation and identifies MGL-1 as an important receptor for cross-presentation of Le^x^-modified antigens. The results presented here may have significant impact in future vaccine design and may also provide insight into mechanisms that trigger recognition of self-antigens in autoimmune disease.

Essential revisions:

This paper addresses an important question and is well written, presented and discussed. The experiments are generally well designed, controlled and conducted. However, there are several substantive concerns raised by the reviewers and the authors are encouraged to address them.

1) Questions arose regarding the identity of the compartment of cross-presentation. Using kinetics experiments, the authors claim that Ova-Le^x^ re-routes antigen from an early EEA1^+^Rab11^+^ endosome to an LAMP1^+^Rab11^+^ endolysosomal compartment where antigen lasts for long periods of time. To formally prove that Le^x^ modification results in re-routing of antigen, the authors should include the routing kinetics of native Ova.

2) As presented in Figure 3, the results neither prove nor suggest whether cross-presentation occurs in EA1^+^Rab11^+^ endosome, the LAMP1^+^Rab11^+^ endolysosomal compartment or both. It would strengthen the manuscript substantially to provide some functional evidence for cross-presentation in these compartments.

3) It is particularly important that the authors demonstrate by confocal imaging the co-localization of Ovalbumin Le^x^ in Rab11^+^Lamp1^+^ compartments in WT and MGL-1^-/-^ BMDCs, as the resolution of the FACS based immuno-fluorescence analyses is less convincing.

4) It is also critical that the authors compare MGL-1-Fc with a Mannose receptor Fc fusion protein and recognition of Ova.

5) Apart from performing these additional experiments in support of the study claims, the authors need to re-analyze several results, e.g., Figure 2, Figure 3, Figure 4, etc., and in other instances show representative flow cytometry data.

---

## [Author Response]

Essential revisions:

*1) Questions arose regarding the identity of the compartment of cross-presentation. Using kinetics experiments, the authors claim that Ova-Le^x^ re-routes antigen from an early EEA1^+^Rab11^+^ endosome to an LAMP1^+^Rab11^+^ endolysosomal compartment where antigen lasts for long periods of time. To formally prove that Le^x^ modification results in re-routing of antigen, the authors should include the routing kinetics of native Ova.*

We have performed additional experiments to compare the intracellular routing of OVA-Le^X^ with that of OVA at different time points by confocal laser scan microscopy. Co-staining with EEA-1, LAMP-1 and Rab11 revealed that upon internalization, OVA-Le^X^ is rapidly shuttled to EEA-1^+^Rab11^+^ compartments from where it moves to LAMP-1^+^Rab11^+^ compartments after 2 hrs (see Figure 6—figure supplement 3). In contrast, OVA-Le^X^ does not end up in LAMP-1^+^Rab11^+^ compartments in MGL1 KO BM-DCs (Figure 6—figure supplement 3). Importantly, these data confirm our imaging flow cytometric analysis (Figure 6 in the revised manuscript). While OVA also rapidly routes to EEA-1^+^Rab11^+^ compartments, it does not move to LAMP-1^+^Rab11^+^ at later time points, in contrast to OVA-Le^X^ (Figure 6—figure supplement 3). This indicates that the modification of OVA with Le^X^ that binds MGL1 alters the intra-cellular routing of the antigen.

The new confocal data obtained on the differential intracellular routing of OVA-Le^X^ and OVA are added to the manuscript (Results, subsection “Modification of OVA with LeX alters the intracellular routing of OVA” and Figure 6, panel C. Additionally, a more detailed representation of the confocal analysis we performed is shown in Figure 6—figure supplement 3). Furthermore, we have adapted the title of the manuscript to “Glycan modification of antigen alters its intracellular routing in dendritic cells, promoting priming of T cells.”

*2) As presented in Figure 3, the results neither prove nor suggest whether cross-presentation occurs in EA1^+^Rab11^+^ endosome, the LAMP1^+^Rab11^+^ endolysosomal compartment or both. It would strengthen the manuscript substantially to provide some functional evidence for cross-presentation in these compartments.*

Indeed, the reviewers are correct that the data as presented as well our new confocal imaging data do not show that OVA-Le^X^ is cross-presented within Rab11^+^LAMP-1^+^ compartments. We therefore weakened the Discussion as we do not want to claim that cross-presentation is occurring there. To analyze this in depth we would need to generate Rab11-deficient (WT and MGL1 KO) BM-DCs by CRISPR-Cas9 technology as well as further identification of this compartment, which we consider out of the scope of this manuscript.

*3) It is particularly important that the authors demonstrate by confocal imaging the co-localization of Ovalbumin Le^x^ in Rab11^+^Lamp1^+^ compartments in WT and MGL-1^-/-^ BMDCs, as the resolution of the FACS based immuno-fluorescence analyses is less convincing.*

As mentioned above, we have performed additional experiments to analyze the intracellular routing of OVA-Le^X^ at different time points by confocal laser scan microscopy. These new data confirm our imaging flow cytometric analysis (Figure 6+B of the revised manuscript) as they show that upon internalization, OVA-Le^X^ is rapidly shuttled to EEA-1^+^Rab11^+^ compartments from where it moves to LAMP-1^+^Rab11^+^ compartments (see Figure 6—figure supplement 3 below). Moreover, neither OVA nor OVA-Le^X^ in MGL1 KO BM-DCs does end up in LAMP-1^+^Rab11^+^ compartments (Figure 6—figure supplement 3+C). Thus, together these data strongly suggest that OVA modified with Le^X^, and internalized via MGL1 is differently routed intra-cellular. These confocal images are added to the manuscript (Results, subsection “Modification of OVA with LeX alters the intracellular routing of OVA” and Figure 6, panel C. Additionally, a more detailed representation of the confocal analysis is shown in Figure 6—figure supplement 3).

*4) It is also critical that the authors compare MGL-1-Fc with a Mannose receptor Fc fusion protein and recognition of Ova.*

We have performed these additional experiments, and indeed confirm that the MR equally binds to OVA and OVA-Le^X^, while MGL1-Fc specifically binds to OVA-Le^X^ (see Figure 8). These data further strengthen the previous binding of the MR to endogenous glycans on OVA (Burgdorf et al., 2007. Science, 316, 612-616) and show that modification of OVA with Le^X^ increases its affinity to bind MGL1, favoring efficient cross-presentation, independent of MyD88 and Cathepsin-S.

The comparison of MR-Fc with MGL-1-Fc binding to OVA-Le^X^ and OVA has been added to the manuscript (Results, subsection “Identification of glycans on native and glycan-modified OVA, and the consequences for CLR-specific binding and cross-presentation” and Figure 1, panel B).

Author response image 1.Generation of OVA-neo-glycoconjugates with LeX that confers binding of OVA to MGL1.(**A**) ELISA showing functional modification of OVA with LeX glycans, resulting in binding of MGL1-Fc. Unconjugated OVA does not carry any ligands for MGL1. Modification of OVA with LeX did not alter the ability to bind to MR as illustrated by equal binding kinetics of MR-Fc to OVA and OVA-LeX (**B**). (C+D) Linear regression analysis of MGL1-Fc and MR-Fc binding to OVA and OVA-LeX at the concentrations used in the study, clearly shows differential binding of OVA-LeX to MGL1 but not to MR due to the modification with LeX.**DOI:**
http://dx.doi.org/10.7554/eLife.11765.023

5) Apart from performing these additional experiments in support of the study claims, the authors need to re-analyze several results, e.g., Figure 2, Figure 3, Figure 4, etc., and in other instances show representative flow cytometry data.

We have re-analyzed the data shown in Figure 2, Figure 3 and Figure 4 of the manuscript. Based on this re-analysis only the scatter plot showing the frequency of IFN-γ-producing CD8^+^ T cells has been adapted (Figure 2). Representative pictures of flow cytometric analysis as well as adapted scatter plot have been added to the manuscript (Figure 2—figure supplement 1 and Figure 2—figure supplement 2, Figure 3—figure supplement 2, Figure 4—figure supplement 1 and Figure 4—figure supplement 3).